# Asymmetric Synthesis and Cytotoxicity Evaluation of Right-Half Models of Antitumor Renieramycin Marine Natural Products

**DOI:** 10.3390/md17010003

**Published:** 2018-12-20

**Authors:** Takehiro Matsubara, Masashi Yokoya, Natchanun Sirimangkalakitti, Naoki Saito

**Affiliations:** Graduate School of Pharmaceutical Sciences, Meiji Pharmaceutical University, 2-522-1 Noshio, Kiyose, Tokyo 204-8588, Japan; m176239@std.my-pharm.ac.jp (T.M.); kikns@hotmail.com (N.S.)

**Keywords:** renieramycin, 1,2,3,4-tetrahydroisoquinoline, cytotoxicity, asymmetric synthesis, marine natural product

## Abstract

A general protocol for the asymmetric synthesis of 3-*N*-arylmethylated right-half model compounds of renieramycins was developed, which enabled structure–activity relationship (SAR) study of several 3-*N*-arylmethyl derivatives. The most active compound (**6a**) showed significant cytotoxic activity against human prostate cancer DU145 and colorectal cancer HCT116 cell lines (IC_50_ = 11.9, and 12.5 nM, respectively).

## 1. Introduction

Natural products belonging to the *bis*-1,2,3,4-tetrahydroisoquinoline family, such as renieramycins, saframycins, and ecteinascidins, have attracted considerable attention due to their potent biological activities, structural diversity, and meager availability in nature (Figure 1) [1]. We have discovered a number of renieramycin marine natural products having extraordinary structures from blue sponges collected in Thailand and the Philippines [2,3,4]. For example, renieramycin M (**1m**) isolated from the Thai blue sponge *Xestospongia* sp. has *p*-quinones in both terminal rings [2]. In contrast, renieramycins T (**1t**) and U (**1u**) share a common A-ring with ecteinascidin 743 (ET-743, **2**), which has already been approved as an anticancer agent [3]. In addition, the A-ring of renieramycin Y (**1y**) has the same substituent pattern as the E-ring of **2** [4]. These renieramycins have similar structures to **2** and are expected to have similar potent antitumor activity. However, the amount obtainable from nature is scarce, and this has set back the implementation of detailed biological tests.

Under these circumstances, we have been developing a total synthesis of these fascinating marine natural products. We have succeeded in the total syntheses of renieramycins G–I, cribrostatin 4, and renieramycin T [5,6,7,8]. However, the long and tedious procedures for the total synthesis of these natural products have impeded detailed structure–activity relationship (SAR) studies.

## 2. Results

We have been trying to simplify the structures of renieramycins without impairing their biological activities. Several right-hand (CDE-ring system) and left-hand (ABC-ring system) model compounds were prepared, and their in vitro cytotoxic activities against human cancer cell lines were tested [9,10]. These efforts have yielded CDE-ring model compounds (±)-**3** to (±)-**6a** (Figure 2), and the presence of amino nitriles was found to induce at nanomolar concentrations (Table 1) [9]. 3-*N*-benzyl derivative (±)-**6a** exhibited approximately five and nine times more potent cytotoxic activity against HCT116 and QG56, respectively, than 3-*N*-methylated derivative (±)-**5**, indicating the importance of the substituent at 3-nitrogen. 

From the structure comparison of (±)-**6a** and **1m**, we expected that 3-*N*-Bn would correspond to the A-ring of **1m**. Thus, **6b** having an arylmethyl group whose substituent pattern was similar to the A-ring of **1m**, and **6c** having the same A-ring as **1y** were set as the new target molecules (Figure 3). 

A summary of our previously reported synthesis of racemic **6a** is shown in Scheme 1 [9,11]. Conversion of 1,2,4-Trimethoxybenzene (**7**) into piperadine-2-5-dione derivative **8** took seven steps, and treatment of **8** with NaH and BnBr gave *N*-benzyl compound **9**. Racemic compound **6a** was prepared from **9** in ten steps. As the 3-*N*-arylmethyl group was critical to generate strong antitumor activity, derivatives with different 3-*N*-arylmethyl groups were prepared in subsequent steps. In addition, it is very interesting to compare the biological activities of the racemic form and the optically active form [12]. Thus, in order to facilitate the synthesis of structural analogs, a new asymmetric synthetic route for preparing the 3-*N*-arylmethyl group in the later steps should be developed. 

An outline of an alternative synthetic strategy to facilitate the asymmetric synthesis of various 3-*N*-arylmethyl derivatives is shown in Figure 4. We envisioned that the final step in the asymmetric synthesis of **6** should involve a reductive cyanation of the lactam carbonyl followed by a two-step oxidation of the phenol into *p*-quinone from **10**. An *N*-arylmethyl group, which would be important for the cytotoxic activity, should be installed on lactam **11**. The C-ring formation proceeded automatically from the lactonization of the primary amine, which was generated by the deprotection of the *N*-Cbz protecting group of **12**. The synthesis of 1,3-*cis*-1,2,3,4-tetrahydroisoquinoline **12** was accomplished via the regio- and diastereoselective Pictet–Spengler cyclization reaction of aminophenol (−)-**13** with *N*-Cbz glyoxal **14** [13]. Starting material (−)-**13** was easily prepared from l-tyrosine according to Liu’s method [14]. 

Our synthesis began with the highly regio- and diastereoselective Pictet–Spengler cyclization reaction of aminophenol (−)-**13** with *N*-protected glyoxal **14**, which was prepared from commercially available 2-(Cbz-amino)-1-ethanol according to a previously reported oxidation reaction [13] (Scheme 2). This cyclization reaction of (−)-**13** with **14** in CH_2_Cl_2_ and 2,2,2-trifluoroethanol at −40 °C for 5 h provided (1*R*,3*S*)-1,2,3,4-tetrahydroisoqunoline (−)-**12** in 82% yield. The ^1^H-NMR spectrum of (−)-**12** was complicated due to the presence of carbamate rotational isomers; thus, the structure determination of (−)-**12** was completed after the transformation of (−)-**12** into tricyclic compound (−)-**16**. 1,2,3,4-Tetrahydroisoquinoline (−)-**12** was de-protected by catalytic hydrogenation to afford primary amine **15**, which spontaneously intramolecularly cyclized to tricyclic compound (−)-**16** in 84% yield. It was confirmed that (−)-**12** obtained by the Pictet–Spengler reaction had the desired 1,3-*cis* configuration. Reductive amination of (−)-**16** followed by *N*-methylation provided (−)-**17** in 50% yield along with overreacted compound (−)-**18** in 13% yield. Obtained side product (−)-**18** was easily converted into (−)-**17** in 82% yield by treatment with ammonia in methanol. 

The alkylation of (−)-**17** with benzyl bromide in the presence of 10 equivalents of sodium hydride afforded dibenzylated product (−)-**19** in 97% yield (Scheme 3). The lactam carbonyl of (−)-**19** was partially reduced with LiAlH_2_(OEt)_2_ [15] in tetrahydrofuran (THF) to generate the aminal, which was treated with KCN and water to provide α-aminonitrile (−)-**20** in 74% yield as a single diastereomer. Chemoselective *O*-debenzylation was achieved with BCl_3_ in the presence of pentamethylbenzene to give desired phenol (−)-**21** in 89% yield [16]. Finally, oxidation of (−)-**21** with O_2_ in the presence of salcomine afforded chiral 3-*N*-benzylated CDE-ring model compound (−)-**6a**. (−)-**6a** was confirmed to have 99%ee by high performance liquid chromatography (HPLC) analysis, proving that the chiral center in l-tyrosine did not cause any racemization in this synthetic route. 

With tricyclic chiral model (−)-**6a** in hand, **6d** having an arylmethyl group with more electron-rich trimethoxy substituents was prepared (Scheme 4). The phenol of (−)-**17** was selectively protected with 1.2 equivalents each of NaH and BnBr in *N*,*N*-dimethylformamide (DMF) to give (−)-**11**, which could be used to prepare several kinds of 3-*N*-alkylated compounds. The reaction of (−)-**11** with substituted benzyl bromide **22**, which was obtained by a reported method [17], produced 3-*N*-arylmethylated (−)-**23** in 55% yield. The conversion (−)-**23** into (−)-**25** was carried out using a similar three-step sequence to that shown above, and salcomine oxidation of (−)-**25** gave *p*-quinone (+)-**6d** in 79% yield. 

The preparation of right-half model compounds **6b** and **6c** whose A-ring substitution patterns correspond to those of **1m** and **1y**, respectively, was carried out as follows (Scheme 5). Benzyl bromide **27** was prepared by the Appel reaction of corresponding alcohol **26** [18] in 99% yield. Alkylation of the lactam nitrogen of (−)-**11** with **27** gave (−)-**28** in 72% yield. Reductive cyanation of (−)-**28** generated aminonitrile (−)-**29** in 71% yield. Debenzylation of (−)-**29** by using BCl_3_ gave a crude product that was expected to contain iminium by-products produced by the cyano group elimination. Thus, the crude product without further purification was treated with KCN to furnish desired phenol (−)-**30** in 85% yield. Finally, bisphenol (−)-**30** was oxidized with two equivalents of salcomine in oxygen atmosphere to give (−)-**6b** and (+)-**6c** in 43% and 31% yields, respectively. This oxidation could be controlled by adjusting the proportion of salcomine, as shown in Table 2. 

Although the detailed molecular mechanism underlying the antitumor activities of renieramycin marine natural products were unclear, we had speculated that the cyano or hydroxyl substituent at C-21 position of renieramycin would be essential for the potent cytotoxic activity. Elimination of the functional group at C-21 produced an electrophilic iminium ion species that was implicated in the formation of covalent bonds with DNA [19]. In 2008, Avendaño and co-workers reported a series of 1,2,3,4-tetrahydroisoquinolines with antitumor activities that were attributed to both apoptosis in the G2/M checkpoint and cytostatic activity in the G1 phase [20]. In addition, we prepared a series of renieramycin left-half model compounds from phenylalanine derivatives, and re-confirmed the importance of the C-21 cyano group for favorable activity [10]. Four right-half chiral model compounds **6a**–**6d** and racemic **6a** and **6b**, including natural renieramycin M (**1m**) as positive control, were tested in vitro for cytotoxicity toward two representative human cancer cell lines (prostate cancer DU145 and colorectal cancer HCT116) using the CCK-8 assay (Table 3). Interestingly, the structure of the E-ring was found to be important for the enhanced biological activity. In order to examine the influence of the E-ring on the bioactivity, the IC_50_ values of three compounds (**6a**, **20**, **21**), in which the 3-*N*-Bn substituent and the C-4 cyano group were fixed, were compared. *p*-Quinone **6a** was the most active, phenol **21** had comparable activity to **6a**, and benzyl ether **20** showed markedly decreased activity. Then, the importance of the cyano group at C-4 position was also confirmed in the right-half models. A significant decrease in cytotoxic activity was observed when the lactam carbonyl at C-4 position was converted into an aminonitrile (i.e., conversion of **19** into **20**, **23** into **24**, and **28** into **29**). However, it was interesting that **28** showed moderate activity even though C-4 had a lactam carbonyl. 

Next, on comparing the IC_50_ values of **21**, **25**, and **30** having characteristic 3-*N*-arylmethyl groups, **30** was found to show the least potent activity, whereas **25** with a trimethoxy arylmethyl group exhibited more potent activity. In the case of **21**, which has an unsubstituted arylmethyl group, very strong activity at nanomolar concentrations was observed. A similar tendency was also observed in the model compounds. Among compounds **6a**, **6b**, **6c**, and **6d** whose E-rings were a quinone, phenol **6c**, which has a phenol in the A-ring exhibited the weakest activity, whereas 3-*N*-benzyl **6a** showed the strongest activity. 

Finally, the effect of optical activity on the biological activity was investigated. It was recently confirmed that optically active (−)-**1m** obtained from nature had approximately two to three times stronger activity than racemic (±)-**1m** [21]. Unlike **1m,** the chirality of racemic **6a** and **6b** and their chiral counterparts had no effect on their cytotoxic activities. Worth noting was that **6a** with *p*-quinone on the E-ring, a cyano group at C-4 position, and 3-*N*-benzyl was the most active compound against the two types of cancer cell lines, and had similar potency to natural **1m**.

## 3. Experimental Section

### 3.1. Chemistry

IR spectra were obtained with a Shimadzu IRAffinity-1 FT-IR spectrometer (Shimadzu Corporation, Kyoto, Japan). Optical rotations were measured with Horiba SEPA-500 polarimeters (Horiba Ltd., Kyoto, Japan). ^1^H- and ^13^C-NMR spectra were recorded on a JEOL JNM-AL 400 NMR spectrometer (JEOL Ltd., Tokyo, Japan) at 400 MHz for ^1^H and 100 MHz for ^13^C; and a JEOL JNM-AL 300 NMR spectrometer at 300 MHz for ^1^H and 75 MHz for ^13^C (ppm, *J* in Hz with tetramethylsilane (TMS) as internal standard). All proton and carbon signals were assigned by extensive NMR measurements using correlation spectroscopy (COSY), Heteronuclear Multiple-Bond Correlation (HMBC), and Heteronuclear Multiple Quantum Correlation (HMQC) techniques. Mass spectra were recorded on a JEOL JMS 700 instrument (JEOL Ltd., Tokyo, Japan) with a direct inlet system operating at 70 eV.

#### 3.1.1. Synthesis of 1,2,3,4-Tetrahydroisoquinoline-3-carboxylate (**12**)

To a stirred solution of aldehyde **14** (2.73 g, 14.1 mmol, 1.3 eq.) and 4 Å molecular sieves (2.60 g) in CH_2_Cl_2_ (70 mL), a solution of amine **13** (2.60 g, 10.9 mmol), acetic acid (160 µL) and 2,2,2-trifluoroethanol (10 mL) was added slowly over 6 min at −40 °C. After being stirred at −40 °C for 5 h, the reaction mixture was neutralized with NaHCO_3_, and then filtered through a Celite pad. The filtrate was concentrated under reduced pressure and the residue was purified by column chromatography (CHCl_3_–EtOAc = 2:1) to afford compound **12** (3.71 g, 82%) as a pale yellow amorphous. [α]${}_{D}^{24}$ −83.9 (*c* 1.1, CHCl_3_); ^1^H NMR (400 MHz, C_5_D_5_N, 80 °C) δ 10.30 (1H, brs, NH or OH), 7.46–7.25 (5H, m, Bn-H), 6.54 (1H, s, 5-H), 5.35–5.25 (2H, m, 5’-H), 4.90 (1H, br t, *J* = 3.2 Hz, 1-H), 4.31–4.27 (1H, m, Bn-H), 4.00–3.94 (1H, m, Bn-H), 3.79–3.75 (1H, m 3-H), 3.73 (3H, s, 7-OCH_3_), 3.70 (3H, s, 3-COOCH_3_), 3.02 (2H, brd, *J* = 6.4 Hz, 4-H), 2.30 (3H, s, 6-CH_3_); ^13^C-NMR (400 MHz, C_5_D_5_N, 80 °C) δ174.0 (s, COOCH_3_), 157.6 (s, C-3’), 148.2 (s, C-8), 145.9 (s, C-7), 138.3 (s, Bn), 132.2 (s, C-4a), 129.5 (s, C-6), 128.8 (d, Bn), 128.2 (d, Bn), 128.0 (d, Bn), 122.7 (s, C-8a), 121.9 (d, C-5), 66.4 (t, C-5’), 60.2 (q, 7-OCH_3_), 56.0 (d, C-3), 54.2 (d, C-1), 51.8 (q, 3-COOCH_3_), 46.8 (t, C-1’), 33.8 (t, C-4), 15.8 (q, 6-CH_3_); IR (CHCl_3_) 3520, 3437, 3024, 3015, 2955, 2359, 2342, 1717, 1506, 1456, 1233, 1059 cm^−1^; FABMS *m/z* 415 [M + H]^+^; HRFABMS *m/z* 415.1867 ([M + H]^+^, calcd for C_22_H_27_N_2_O_6_ 415.1869).

#### 3.1.2. Synthesis of (1*R*,5*S*)-10-Hydroxy-9-methoxy-8-methyl-2,3,5,6-tetrahydro-1,5-epiminobenzo[d]azocin-4(1H)-one (**16**)

A solution of **12** (2.96 g, 7.14 mmol) in EtOH (370 mL) was hydrogenated over 10% Pd/C (55%wet, 1.52 g, 1.43 mmol) at 25 °C for 5 h under 3.5 atm hydrogen. The catalyst was removed by filtration and the filtrate was concentrated under reduced pressure and the residue was purified by SiO_2_ flash column chromatography (CHCl_3_−MeOH = 9:1) to afford compound **16** (1.49 g, 84%) as a pale brown solid. [α]${}_{D}^{24}$ −177.0 (*c* 1.0, CHCl_3_); ^1^H-NMR (400 MHz, DMSO-d_6_) δ 8.80 (1H, brs, 10-OH), 7.39 (1H, d, *J* = 4.0 Hz, 3-*N*-H), 6.37 (1H, s, 7-H), 4.19 (1H, d, *J* = 4.4 Hz, 1-H), 3.60 (3H, s, 9-OCH_3_), 3.53 (1H, dd, *J* = 11.2, 4.4 Hz, 2-H), 3.49 (1H, d, *J* = 6.2 Hz, 5-H), 3.07 (1H, dd, *J* = 11.2, 4.0 Hz, 2-H), 2.86 (1H, dd, *J* = 16.5, 6.2 Hz, 6-H), 2.59 (1H, d, *J* = 16.5 Hz, 6-H), 2.13 (3H, s, 8-CH_3_); ^13^C-NMR (100 MHz, DMSO-d_6_) δ 171.4 (s, C-4), 145.7 (s, C-10), 143.7 (s, C-9), 129.7 (s, C-6a), 128.7 (s, C-8), 122.9 (s, C-10a), 120.6 (d, C-7), 59.9 (q, 9-OCH_3_), 52.4 (d, C-5), 47.5 (t, C-2), 43.7 (d, C-1), 32.0 (t, C-6), 15.5 (q, 8-CH_3_); IR (KBr) 3497, 3428, 3345, 3246, 1643, 1335, 1273, 1069, 1001 cm^−1^; EIMS *m/z* (%) 248 (M^+^, 24), 191 (17), 190 (100), 175 (16); HREIMS *m/z* 248.1162 (M^+^, calcd for C_13_H_16_N_2_O_3_ 248.1161).

#### 3.1.3. Synthesis of (1*R*,5*S*)-10-Hydroxy-9-methoxy-8,11-dimethyl-2,3,5,6-tetrahydro-1,5-epiminobenzo[d]azocin-4(1H)-one (**17**) and (1*R*,5*S*)-10-hydroxy-3-(hydroxymethyl)-9-methoxy-8,11-dimethyl-2,3,5,6-tetrahydro-1,5-epiminobenzo[d]azocin-4(1H)-one (**18**)

To a stirred solution of amine **16** (248 mg, 1.00 mmol) in CH_3_CN (34 mL) was added 37% HCHO (1.60 mL, 20.0 mmol, 20 eq.). The reaction mixture was stirred for 15 min, after which NaCNBH_3_ (700 mg, 10.0 mmol, 10 eq.) was added. The reaction mixture was stirred for 15 min, after which AcOH (570 µL, 10.0 mmol, 10 eq.) was added dropwise over 3 min. The reaction mixture was stirred for 5 min, after which 2 N HCl (34 mL) was added 1 portion. The reaction was heated to 60 °C and was stirred for 16 h. The reaction was quenched with saturated NaHCO_3_ (200 mL) and extracted with CHCl_3_−MeOH = 9:1 (3 × 150 mL). The combined extracts were washed with H_2_O (100 mL), brine (100 mL), dried over Na_2_SO_4_, and concentrated in vacuo to give a residue. The residue was purified by SiO_2_ flash column chromatography (benzene−acetone = 1:2) to afford compound **18** (39.0 mg, 13%) as a colorless solid, and with CHCl_3_−MeOH (6:1) to afford **17** (130 mg, 50%) as a colorless solid.

**17**: [α]${}_{D}^{24}$ −224.0 (*c* 1.0, CHCl_3_); ^1^H-NMR (400 MHz, CDCl_3_) δ 6.49 (1H, s, 7-H), 5.92 (1H, brs, 3-*N*-H), 4.17 (1H, d, *J* = 4.5 Hz, 1-H), 3.91 (1H, dd, *J* = 11.6, 4.5 Hz, 2-H), 3.77 (3H, s, 9-OCH_3_), 3.57 (1H, d, *J* = 6.6 Hz, 5-H), 3.30 (1H, ddd, *J* = 11.6, 3.8, 0.9, 2-H), 3.17 (1H, dd, *J* = 17.0, 6.6 Hz, 6-H), 2.79 (1H, d, *J* = 17.0 Hz, 6-H), 2.52 (3H, s, *N*-CH_3_), 2.25 (3H, s, 8-CH_3_); ^13^C-NMR (100 MHz, CDCl_3_) δ 172.3 (s, C-4), 145.5 (s, C-10), 143.3 (s, C-9), 129.3 (s, C-6a), 129.0 (s, C-8), 121.9 (d, C-7), 119.2 (s, C-10a), 60.8 (q, 9-OCH_3_), 59.2 (d, C-5), 50.0 (d, C-1), 45.3 (t, C-2), 40.1 (q, *N*-CH_3_), 27.8 (t, C-6), 15.8 (q, 8-CH_3_); IR (KBr) 3265, 2938, 2874, 1684, 1645, 1495, 1335, 1265, 1055, 1038 cm^−1^; EIMS *m/z* (%) 262 (M^+^, 20), 205 (17), 204 (100), 189 (16); HREIMS *m/z* 262.1317 (M^+^, calcd for C_14_H_18_N_2_O_3_ 262.1317).

**18**: [α]${}_{D}^{24}$ −197.7 (*c* 1.0, CHCl_3_); ^1^H-NMR (400 MHz, CDCl_3_) δ 6.47 (1H, s, 7-H), 4.88 (1H, d, *J* = 10.4 Hz, 3-*N*-CH_2_OH), 4.56 (1H, d, *J* = 10.4 Hz, 3-*N*-CH_2_OH), 4.22 (1H, d, *J* = 4.6 Hz, 1-H), 4.08 (1H, dd, *J* = 11.5, 4.6 Hz, 2-H), 3.76 (3H, s, 9-OCH_3_), 3.61 (1H, d, *J* = 6.5 Hz, 5-H), 3.35 (1H, d, *J* = 11.5, 2-H), 3.15 (1H, dd, *J* = 17.0, 6.5 Hz, 6-H), 2.77 (1H, d, *J* = 17.0 Hz, 6-H), 2.49 (3H, s, 11-*N*-CH_3_), 2.25 (3H, s, 8-CH_3_); ^13^C-NMR (100 MHz, CDCl_3_) δ 172.1 (s, C-4), 145.6 (s, C-10), 143.4 (s, C-9), 129.4 (s, C-6a), 128.6 (s, C-8), 121.7 (d, C-7), 118.9 (s, C-10a), 71.6 (t, C-3), 60.7 (q, 9-OCH_3_), 59.1 (d, C-5), 50.7 (d, C-1), 50.0 (t, C-2), 39.8 (q, *N*-CH_3_), 27.1 (t, C-6), 15.7 (q, 8-CH_3_); IR (KBr) 3489, 3150, 2949, 2934, 1618, 1504, 1236, 1055, 1034 cm^−1^; EIMS *m/z* (%) 292 (M^+^, 3), 262 (17), 205 (18), 204 (100), 189 (16); HREIMS *m/z* 292.1424 (M^+^, calcd for C_15_H_20_N_2_O_4_ 292.1423). 

#### 3.1.4. Synthesis of **17** from **18**

To a stirred solution of lactam **18** (262 mg, 0.896 mmol) in MeOH (26 mL) was added NH_4_OH (10.5 mL) at room temperature (rt). The reaction mixture was stirred for 16 h. The reaction was quenched with conc. HCl at 0 °C, and then neutralized with 5% NaHCO_3_. The reaction mixture was diluted with H_2_O (200 mL) and extracted with CHCl_3_−MeOH = 9:1 (4×50 mL). The combined extracts were washed with brine (100 mL), dried over Na_2_SO_4_, and concentrated in vacuo to give a residue. The residue was purified by SiO_2_ flash column chromatography (CHCl_3_−MeOH = 9:1) to afford compound **17** (19.3 mg, 82%) as a colorless solid.

#### 3.1.5. Synthesis of (1*R*,5*S*)-3-Benzyl-10-(benzyloxy)-9-methoxy-8,11-dimethyl-2,3,5,6-tetrahydro-1,5-epiminobenzo[d]azocin-4(1H)-one (**19**)

To a stirred solution of lactam **17** (10.0 mg, 38.0 µmol) and benzyl bromide (10.0 µL, 76.0 µmol, 2.0 eq.) in DMF (1 mL) was added NaH (60% oil dispersion, 15.2 mg, 381 µmol, 10.0 eq.) at 0 °C. The reaction mixture was stirred at 25 °C for 17 h. The reaction mixture was diluted with H_2_O (5 mL) and extracted with Et_2_O (3 × 10 mL). The combined extracts were dried over Na_2_SO_4_, and concentrated in vacuo to give a residue. The residue was purified by SiO_2_ flash column chromatography (CHCl_3_−MeOH = 99:1) to afford compound **19** (16.4 mg, 97%) as a yellow oil. [α]${}_{D}^{24}$ −77.5 (*c* 1.2, CHCl_3_); ^1^H-NMR (400 MHz, CDCl_3_) δ 7.34–7.22 (5H, m, 10-*O*-Bn-H), 7.13–7.04 (3H, m, 3-*N*-Bn-H), 6.75–6.73 (2H, m, 3-*N*-Bn-H), 6.75 (1H, s, 7-H), 4.98 (1H, d, *J* = 11.5 Hz, 10-OCH_2_Ph), 4.84 (1H, d, *J* = 15.1 Hz, 3-*N*-CH_2_Ph), 4.74 (1H, d, *J* = 11.5 Hz, 10-OCH_2_Ph), 4.10 (1H, d, *J* = 15.1 Hz, 3-*N*-CH_2_Ph), 3.89 (1H, brd, *J* = 4.4 Hz, 1-H), 3.68–3.65 (1H, m, 2-H), 3.68 (1H, d, *J* = 6.3 Hz, 5-H), 3.67 (3H, s, 9-OCH_3_), 3.16 (1H, dd, *J* = 17.1, 6.3 Hz, 6-H), 2.92 (1H, d, *J* = 11.7 Hz, 2-H), 2.86 (1H, d, *J* = 17.1 Hz, 6-H), 2.29 (3H, s, 11-*N*-CH_3_), 2.29 (3H, s, 8-CH_3_); ^13^C-NMR (100 MHz, CDCl_3_) δ 170.2 (s, C-4), 149.3 (s, C-9), 148.3 (s, C-10), 137.3 (s, Bn), 136.3 (s, Bn), 131.2 (d, C-8), 128.4 (d×2, Bn), 128.0 (d×2, Bn), 128.2 (s, C-6a), 127.1 (d, Bn), 126.8 (d, Bn), 126.0 (d, C-7), 125.6 (s, C-10a), 74.1 (t, 10-OCH_2_Ph), 59.9 (q, 9-OCH_3_), 59.3 (d, C-5), 51.5 (d, C-1), 50.5 (t, C-2), 48.5 (t, 3-*N*-CH_2_Ar), 39.7 (q, 11-*N*-CH_3_), 27.4 (t, C-6), 15.6 (q, 8-CH_3_); IR (CHCl_3_) 3009, 2940, 1636, 1493, 1454, 1337, 1059, 698 cm^−1^; EIMS *m/z* (%) 442 (M^+^, 30), 351 (10), 295 (23), 294 (100), 204 (30), 203 (37), 91 (11); HREIMS *m/z* 442.2254 (M^+^, calcd for C_28_H_30_N_2_O_3_ 442.2256). 

#### 3.1.6. Synthesis of (1*R*,4*R*,5*S*)-3-Benzyl-10-(benzyloxy)-9-methoxy-8,11-dimethyl-1,2,3,4,5,6-hexahydro-1,5-epiminobenzo[d]azocine-4-carbonitrile (**20**)

To a solution of lactam **19** (45.7 mg, 103 µmol) in THF (2.5 mL) at 0 °C was slowly added LiAlH_2_(OEt)_2_ (1.0 mol/L in CH_2_Cl_2_, 1.20 mL, 1.20 mmol, 12 eq.) over 10 min. The reaction mixture was stirred at 0 °C for 3 h. The reaction mixture was quenched with AcOH (120 µL, 2.15 mmol, 20.8 eq.), followed by the addition of KCN (40.4 mg, 620 µmol, 6.0 eq.) in H_2_O (1.0 mL), and stirring was continued for 16 h at 25 °C. The reaction mixture was neutralized with 5% NaHCO_3_ solution and diluted with saturated Rochell’s salt aq., and the mixture was stirred for 1.5 h. The reaction mixture was extracted with CHCl_3_ (3 × 30 mL). The combined extracts were washed with brine (30 mL), dried over Na_2_SO_4_, and concentrated in vacuo to give a residue. The residue was purified by SiO_2_ flash column chromatography (n-Hex.−EtOAc = 4:1) to afford compound **20** (33.2 mg, 74%) as a colorless amorphous. [α]${}_{D}^{24}$ −48.4 (*c* 1.5, CHCl_3_); ^1^H-NMR (400 MHz, CDCl_3_) δ 7.36–7.25 (5H, m, 10-*O*-Bn-H), 7.17–7.14 (3H, m, 3-*N*-Bn-H), 6.90–6.88 (2H, m, 3-*N*-Bn-H), 6.69 (1H, s, 7-H), 5.04 (1H, d, *J* = 11.3 Hz, 10-OCH_2_Ph), 4.85 (1H, d, *J* = 11.3 Hz, 10-OCH_2_Ph), 3.91 (1H, brs, 1-H), 3.83 (3H, s, 9-OCH_3_), 3.65 (1H, s, 4-H), 3.52 (2H, s, 3-*N*-CH_2_Ph), 3.21 (1H, d, *J* = 7.6 Hz, 5-H), 3.02 (1H, dd, *J* = 17.6, 7.6 Hz, 6-H), 2.81 (1H, dd, *J* = 11.2, 3.0 Hz, 2-H), 2.51 (1H, brd, *J* = 11.2, Hz, 2-H), 2.35 (1H, d, *J* = 17.6 Hz, 6-H), 2.32 (3H, s, 8-CH_3_), 2.15 (3H, s, 11-*N*-CH_3_); ^13^C-NMR (100 MHz, CDCl_3_) δ 148.8 (s, C-9), 148.3 (s, C-10), 137.4 (s, Bn), 137.0 (s, Bn), 130.2 (s, C-6a), 130.0 (s, C-8), 128.4 (d×2, Bn), 128.3 (d, Bn), 128.1 (d×2, Bn), 127.2 (d, Bn), 126.5 (s, C-10a), 124.2 (d, C-7), 116.5 (s, 4-CN), 74.4 (t, 10-OCH_2_Ph), 60.0 (q, 9-OCH_3_), 59.1 (d, C-4), 58.9 (d, 3-*N*-CH_2_Ph), 55.4 (d, C-5), 53.4 (t, C-2), 52.8 (d, C-1), 41.2 (q, 11*N*-CH_3_), 25.0 (t, C-6), 15.8 (q, 8-CH_3_); IR (CHCl_3_) 3015, 2936, 2826, 2359, 2342, 2226, 1321, 1227, 1061, 1028, 700 cm^−1^; EI-MS *m/z* (%) 453 (M^+^, 2), 295 (27), 294 (100), 204 (21), 203 (20); HREIMS *m/z* 453.2416 (M^+^, calcd for C_29_H_31_N_3_O_2_ 453.2416).

#### 3.1.7. Synthesis of (1*R*,4*R*,5*S*)-3-Benzyl-10-hydroxy-9-methoxy-8,11-dimethyl-1,2,3,4,5,6-hexahydro-1,5-epiminobenzo[d]azocine-4-carbonitrile (**21**)

To a solution of **20** (20.0 mg, 44.1 μmol) and pentamethylbenzene (65.4 mg, 441 μmol, 10.0 eq.) in CH_2_Cl_2_ (6.0 mL) was added BCl_3_ (1.0 mol/L in CH_2_Cl_2_, 220 µL, 220 µmol, 5 eq.) at −78 °C and the mixture was stirred for 2 h. The reaction mixture was diluted with CH_2_Cl_2_ (5.0 mL) and quenched with saturated NaHCO_3_ solution at 0 °C. The mixture was extracted with CH_2_Cl_2_ (3 × 25 mL). The combined extracts were dried over Na_2_SO_4_ and concentrated in vacuo to give a residue. The residue was purified by SiO_2_ flash column chromatography (n-Hex.−EtOAc = 2:1) to afford compound **21** (14.3 mg, 89%) as a colorless amorphous. [α]${}_{D}^{24}$ −127.4 (*c* 1.5, CHCl_3_); ^1^H-NMR (400 MHz, CDCl_3_) δ 7.19–7.15 (3H, m, 3-*N*-Bn-H), 6.96–6.92 (2H, m, 3-*N*-Bn-H) 6.48 (1H, s, 7-H), 5.67 (1H, brs, 10-OH), 4.09 (1H, brs, 1-H), 3.78 (3H, s, 9-OCH_3_), 3.65 (1H, s, 4-H), 3.62 (1H, d, *J* = 7.8 Hz, 3-*N*-CH_2_Ph), 3.54 (1H, d, *J* = 7.8 Hz, 3-*N*-CH_2_Ph), 3.27 (1H, brd, *J* = 7.5 Hz, 5-H), 3.06 (1H, dd, *J* = 17.6, 7.5 Hz, 6-H), 2.96 (1H, dd, *J* = 11.2, 2.9 Hz, 2-H), 2.72 (1H, d, *J* = 11.2 Hz, 2-H), 2.38 (3H, s, 11-*N*-CH_3_), 2.31 (3H, s, 8-CH_3_), 2.38-2.17 (1H, m, overlapped, 6-H); ^13^C-NMR (100 MHz, CDCl_3_) δ 145.4 (s, C-10), 142.8 (s, C-9), 137.1 (s, Bn), 130.8 (s, C-6a), 128.4 (d, Bn), 128.3 (d, Bn), 128.3 (s, C-8), 127.3 (s, Bn), 120.4 (d, C-7), 119.4 (s, C-10a), 116.6 (4-CN), 60.8 (q, 9-OCH_3_), 59.0 (t, 3-*N*-CH_2_Ph), 58.6 (d, C-4), 55.4 (d, C-5), 53.0 (t, C-2), 52.5 (d, C-1), 41.5 (q, 11-*N*-CH_3_), 25.1 (t, C-6), 15.8 (q, 8-CH_3_); IR (CHCl_3_) 3534, 3019, 2928, 2359, 1454, 1418, 1227, 1059, 1026 cm^−1^; EIMS *m/z* (%) 363 (M^+^, 2), 205 (23), 204 (100), 189 (10); HREIMS *m/z* 363.1943 (M^+^, calcd for C_22_H_25_N_3_O_2_ 363.1947). 

#### 3.1.8. Synthesis of (1*R*,4*R*,5*S*)-3-Benzyl-9-methoxy-8,11-dimethyl-7,10-dioxo-1,2,3,4,5,6,7,10-octahydro-1,5-epiminobenzo[d]azocine-4-carbonitrile (**6a**)

To a solution of phenol **21** (10.0 mg, 27.5 μmol) in THF (1 mL) was added salcomine (8.90 mg, 27.5 μmol, 1.0 eq.) at 25 °C, and the reaction mixture was stirred for 2.5 h under O_2_ atmosphere. The reaction mixture was filtered through a cellulose pad and washed with EtOAc. The filtrate was concentrated in vacuo to give a residue. The residue was purified by SiO_2_ flash column chromatography (CH_2_Cl_2_−MeOH = 99:1) to afford compound **6a** (8.20 mg, 79%) as a dark red amorphous. 99%ee. The ee value was determined by HPLC analysis using CHIRALPAK IC [hexane/EtOH = 80/20, flow 1.0 mL/min, *t*_r_ (minor) = 7.08 min, *t*_r_ (major) = 7.77 min]; [α]${}_{D}^{27}$ −38.0 (*c* 0.3, CHCl_3_); ^1^H-NMR (300 MHz, CDCl_3_) δ 7.25–7.13 (5H, m, 3-*N*-Bn-H), 4.01 (3H, s, 9-OCH_3_), 3.87 (1H, brs, 1-H), 3.66 (1H, d, *J* = 13.2 Hz, 3-*N*-CH_2_Ph), 3.54 (1H, d, *J* = 13.2 Hz, 3-*N*-CH_2_Ph), 3.54 (1H, d, *J* = 2.0 Hz, 4-H), 3.27 (1H, brd, *J* = 7.4 Hz, 5-H), 2.95 (1H, dd, *J* = 11.6, 3.2 Hz, 2-H), 2.70 (1H, dd, *J* = 20.5, 7.4 Hz, 6-H), 2.58 (1H, d, *J* = 11.6 Hz, 2-H), 2.32 (3H, s, 11-*N*-CH_3_), 2.10 (1H, d, *J* = 20.5 Hz, 6-H), 2.01 (3H, s, 8-CH_3_); ^13^C-NMR (100 MHz, CDCl_3_) δ 186.9 (s, C-7), 182.3 (s, C-10), 155.4 (s, C-9), 141.0 (s, C-6a), 137.4 (C-10a), 136.2 (s, Bn), 128.7 (d, Bn), 128.6 (d, Bn), 128.6 (s, C-8), 127.9 (d, Bn), 115.8 (s, 4-CN), 61.0 (q, 9-OCH_3_), 58.9 (t, C-12), 57.7 (d, C-4), 54.5 (d, C-5), 51.7 (t, C-2), 51.3 (d, C-1), 41.5 (q, ^11^*N*-CH_3_), 20.8 (t, C-6), 8.7 (q, 8-CH_3_); IR (CHCl_3_) 3024, 2928, 2855, 2384, 2228, 1653, 1308, 1234, 1155, 1024 cm^−1^; EIMS *m/z* (%) 377 (M^+^, 12), 220 (18), 219 (100), 218 (99), 204 (29), 176 (13), 91 (21); HREIMS *m/z* 377.1737 (M^+^, calcd for C_22_H_23_N_3_O_3_ 377.1739). 

#### 3.1.9. Synthesis of (1*R*,5*S*)-10-(Benzyloxy)-9-methoxy-8,11-dimethyl-2,3,5,6-tetrahydro-1,5-epiminobenzo[d]azocin-4(1H)-one (**11**)

To a solution of lactam **17** (3.17 g, 12.0 mmol) in DMF (250 mL) was slowly added NaH (60% oil dispersion, 580 mg, 15.0 mmol, 1.2 eq.) over 10 min at 0 °C. The reaction mixture was stirred for 30 min at 0 °C, after which BnBr (1.70 mL, 15.0 mmol, 1.2 eq.) was added dropwise over 25 min. The reaction mixture was stirred for 1 h at 25 °C. The reaction mixture was diluted with H_2_O (300 mL) and extracted with CHCl_3_ (3 × 200 mL). The combined extracts were washed with brine (200 mL), dried over Na_2_SO_4_, and concentrated in vacuo to give a residue. The residue was purified by SiO_2_ flash column chromatography (CHCl_3_−MeOH = 99:1) to afford compound **11** (3.81 g, 89%) as a colorless amorphous. [α]${}_{D}^{24}$ −108.2 (*c* 1.1, CHCl_3_); ^1^H-NMR (400 MHz, CDCl_3_) δ 7.41–7.31 (5H, m, 10-O-Bn-H), 6.70 (1H, s, 7-H), 6.18 (1H, brs, 3-*N*-H), 5.18 (1H, d, *J* = 11.6 Hz, 10-OCH_2_Ph), 5.08 (1H, d, *J* = 11.6 Hz, 10-OCH_2_Ph), 3.92 (1H, d, *J* = 4.7 Hz, 1-H), 3.82 (3H, s, 9-OCH_3_), 3.81 (1H, dd, *J* = 10.3, 4.7 Hz 2-H), 3.52 (1H, d, *J* = 6.6 Hz, 5-H), 3.17 (1H, ddd, *J* = 10.3, 3.9, 1.0 Hz, 2-H), 3.13 (1H, dd, *J* = 17.3, 6.6 Hz, 6-H), 2.76 (1H, d, *J* = 17.3 Hz, 6-H), 2.31 (3H, s, 11-*N*-CH_3_), 2.25 (3H, s, 8-CH_3_); ^13^C-NMR (100 MHz, CDCl_3_) δ 172.2 (s, C-4), 149.4 (s, C-9), 148.3 (s, C-10), 137.6 (s, C-1’), 131.5 (s, C-8), 128.6 (d, C-3’, C-5’), 128.4 (d, C-6a), 128.1 (d, C-4’), 128.0 (d, C-2’, C-6’), 126.2 (s, C-10a), 125.8 (d, C-7), 74.2 (t, 10-OCH_2_Ph), 60.1 (q, 9-OCH_3_), 59.0 (d, C-5), 50.5 (d, C-1), 46.3 (t, C-2), 39.8 (q, 11-*N*-CH_3_), 27.4 (t, C-6), 15.8 (q, 8-CH_3_); IR (KBr) 3169, 3028, 2936, 1678, 1337, 1310, 1055, 702 cm^−1^; EIMS *m/z* (%) 352 (M^+^, 38), 295 (23), 294 (100), 261 (14), 204 (46), 203 (61), 174 (10), 91 (11); HREIMS *m/z* 352.1785 (M^+^, calcd for C_21_H_24_N_2_O_3_ 352.1787).

#### 3.1.10. Synthesis of (1*R*,5*S*)-10-(Benzyloxy)-9-methoxy-8,11-dimethyl-3-(2,4,5-trimethoxy-3-methylbenzyl)-2,3,5,6-tetrahydro-1,5-epiminobenzo[d]azocin-4(1H)-one (**23**)

To a solution of NaH (60% oil dispersion, 80.3 mg, 2.00 mmol) in THF (10 mL) was added a solution of lactam **11** (705 mg, 2.00 mmol) in THF (10 mL) at 0 °C. The reaction mixture was stirred for 30 min at 0 °C, after which a solution of bromide **22** (550 mg, 2.00 mmol) in THF (10 mL) was added at 25 °C. The reaction mixture was stirred for 19 h at 25 °C. The reaction mixture was diluted with H_2_O (100 mL) and extracted with CHCl_3_ (3 × 100 mL). The combined extracts were washed with brine (150 mL), dried over Na_2_SO_4_, and concentrated in vacuo to give a residue. The residue was purified by SiO_2_ flash column chromatography (CHCl_3_−MeOH = 49:1) to afford compound **23** (602 mg, 55%) as a yellow gummy solid and starting material **11** (139 mg, 20% recovery). [α]${}_{D}^{27}$ −55.6 (*c* 1.1, CHCl_3_); ^1^H-NMR (400 MHz, CDCl_3_) δ 7.35–7.29 (3H, m, 10-*O*-Bn-H), 7.25–7.22 (2H, m, 10-*O*-Bn-H), 6.72 (1H, s, 7-H), 5.91 (1H, s, 6’-H), 5.06 (1H, d, *J* = 11.3 Hz, 10-OCH_2_Ph), 5.00 (1H, d, *J* = 15.1 Hz, 3-*N*-CH_2_Ar), 4.66 (1H, d, *J* = 11.3 Hz, 10-OCH_2_Ph), 4.14 (1H, d, *J* = 15.1 Hz, 3-*N*-CH_2_Ar), 3.93 (1H, brd, *J* = 4.8 Hz, 1-H), 3.71 (3H, s, 9-OCH_3_), 3.69-3.65 (2H, m, 2-H, 5-H), 3.67 (3H, s, 4’-OCH_3_), 3.57 (3H, s, 2’-OCH_3_), 3.35 (3H, s, 5’-OCH_3_), 3.18 (1H, dd, *J* = 17.2, 6.4 Hz, 6-H), 2.96 (1H, dd, *J* = 11.9 Hz, 2-H), 2.88 (1H, d, *J* = 17.2 Hz, 6-H), 2.32 (3H, s, 11-*N*-CH_3_), 2.25 (3H, s, 8-CH_3_), 2.13 (3H, s, 3’-CH_3_); ^13^C-NMR (100 MHz, CDCl_3_) δ 170.6 (s, C-4), 150.6 (s, C-2’), 149.4 (s, C-9), 149.4 (s, C-5’), 148.6 (s, C-10), 146.8 (s, C-4’), 137.3 (s, Bn), 131.4 (s, C-8), 128.5 (s, C-6a), 128.5 (d, Bn), 128.4 (d, Bn), 128.1 (d, Bn), 126.4 (s, C-10a), 125.4 (d, C-7), 125.1 (s, C-3’), 124.2 (s, C-1’), 107.8 (d, C-6’), 74.1 (t, 10-OCH_2_Ph), 60.9 (q, 2’-OCH_3_), 60.1 (q, 4’-OCH_3_), 59.9 (q, 9-OCH_3_), 59.3 (d, C-5), 55.1 (q, 5’-OCH_3_), 51.4 (d, C-1), 50.5 (t, C-2), 42.6 (t, 3-*N*-CH_2_Ar), 39.7 (q, 11-*N*-CH_3_), 27.5 (t, C-6), 15.7 (q, 8-CH_3_), 9.4 (q, 3’-CH_3_); IR (CHCl_3_) 3024, 2943, 2467, 1641, 1452, 1339, 1244, 1061 cm^−1^; EIMS *m/z* (%) 547 (11), 546 (M^+^, 32), 351 (11), 295 (25), 294 (100), 204 (27), 203 (21), 195 (18); HREIMS *m/z* 546.2731 (M^+^, calcd for C_32_H_38_N_2_O_6_ 546.2730). 

#### 3.1.11. Synthesis of (1*R*,4*R*,5*S*)-10-(Benzyloxy)-9-methoxy-8,11-dimethyl-3-(2,4,5-trimethoxy-3-methylbenzyl)-1,2,3,4,5,6-hexahydro-1,5-epiminobenzo[d]azocine-4-carbonitrile (**24**)

To a solution of lactam **23** (50.0 mg, 92.0 µmol) in THF (3.0 mL) at 0 °C was slowly added LiAlH_2_(OEt)_2_ (1.0 mol/L in CH_2_Cl_2_, 1.10 mL, 1.10 mmol, 12 eq.) over 10 min. The reaction mixture was stirred at 0 °C for 3 h. The reaction mixture was quenched with AcOH (100 µL, 1.90 mmol, 20.8 eq.), followed by the addition of KCN (35.8 mg, 549 µmol, 6.0 eq.) in H_2_O (2.0 mL), and stirring was continued for 14 h at 25 °C. The reaction mixture was neutralized with 5% NaHCO_3_ solution and diluted with saturated Rochell’s salt aq., and the mixture was stirred for 1 h. The reaction mixture was extracted with CHCl_3_ (3 × 30 mL). The combined extracts were washed with brine (40 mL), dried over Na_2_SO_4_, and concentrated in vacuo to give a residue. The residue was purified by SiO_2_ flash column chromatography (n-Hex.−EtOAc = 2:1) to afford compound **24** (38.2 mg, 75%) as a colorless gummy solid. [α]${}_{D}^{27}$ −23.1 (*c* 1.3, CHCl_3_); ^1^H-NMR (400 MHz, CDCl_3_) δ 7.34–7.27 (5H, m, 10-O-Bn-H), 6.59 (1H, s, 7-H), 6.33 (1H, s, 6’-H), 5.07 (1H, d, *J* = 11.3 Hz, 10-OCH_2_Ph), 4.83 (1H, d, *J* = 11.3 Hz, 10-OCH_2_Ph), 3.93 (1H, brs, 1-H), 3.81 (3H, s, 9-OCH_3_), 3.76 (1H, brs, 4-H), 3.71 (3H, s, 4’-OCH_3_), 3.58 (1H, d, *J* = 13.5 Hz, 3-*N*-CH_2_Ar), 3.54 (3H, s, 5’-OCH_3_), 3.46 (1H, d, *J* = 13.5 Hz, 3-*N*-CH_2_Ar), 3.34 (3H, s, 2’-OCH_3_), 3.26 (1H, brd, *J* = 7.7 Hz, 5-H), 3.03 (1H, dd, *J* = 17.9, 7.7 Hz, 6-H), 2.84 (1H, dd, *J* = 10.4, 3.0 Hz, 2-H), 2.56 (1H, d, *J* = 10.4 Hz, 2-H), 2.35 (1H, d, *J* = 17.9 Hz, 6-H), 2.23 (3H, s, 8-CH_3_), 2.15 (3H, s, 11-*N*-CH_3_), 2.13 (3H, s, 3’-CH_3_); ^13^C-NMR (100 MHz, CDCl_3_) δ 150.9 (s, C-2’), 148.8 (s, C-9), 148.7 (s, C-5’), 148.2 (s, C-10), 146.8 (s, C-4’), 137.2 (s, Bn), 129.9 (s, C-6a), 129.8 (s, C-8), 128.3 (d, Bn), 128.2 (d, Bn), 127.9 (d, Bn), 126.7 (s, C-10a), 125.4 (s, C-3’), 124.4 (s, C-1’), 124.0 (d, C-7), 116.5 (s, 4-CN), 109.7 (d, C-6’), 74.2 (t, 10-OCH_2_Ph), 60.7 (q, 2’-OCH_3_), 59.9 (q, 4’-OCH_3_), 59.8 (q, 9-OCH_3_), 59.0 (d, C-4), 55.2 (d, C-5), 55.1 (q, 5’-OCH_3_), 53.7 (t, C-2), 53.1 (t, 3-*N*-CH_2_Ar), 52.6 (d, C-1), 41.0 (q, 11-*N*-CH_3_), 24.9 (t, C-6), 15.5 (q, 8-CH_3_), 9.2 (q, 3’-CH_3_); IR (CHCl_3_) 3015, 2938, 2226, 1485, 1321, 1227, 1088, 1011 cm^−1^; EIMS *m/z* (%) 557 (M^+^, 1), 295 (25), 294 (100), 204 (13), 203 (16); HREIMS *m/z* 557.2893 (M^+^, calcd for C_33_H_39_N_3_O_5_ 557.2890). 

#### 3.1.12. Synthesis of (1*R*,4*R*,5*S*)-10-Hydroxy-9-methoxy-8,11-dimethyl-3-(2,4,5-trimethoxy-3-methylbenzyl)-1,2,3,4,5,6-hexahydro-1,5-epiminobenzo[d]azocine-4-carbonitrile (**25**)

To a solution of **24** (115 mg, 206 µmol) and pentamethylbenzene (306 mg, 2.06 mmol, 10 eq.) in CH_2_Cl_2_ (30 mL) was added BCl_3_ (1.0 mol/L in CH_2_Cl_2_, 1.00 mL, 1.00 mmol, 5 eq.) over 10 min at −78 °C and the mixture was stirred for 2 h. The reaction mixture was diluted with CH_2_Cl_2_ (20 mL) and quenched with saturated NaHCO_3_ solution at 0 °C. The mixture was extracted with CH_2_Cl_2_ (3 × 100 mL). The combined extracts were dried over Na_2_SO_4_ and concentrated in vacuo to give a residue. The residue was purified by SiO_2_ flash column chromatography (n-Hex.−EtOAc = 2:1) to afford compound **25** (80.7 mg, 84%) as a colorless amorphous. [α]${}_{D}^{26}$ −35.9 (*c* 0.9, CHCl_3_); ^1^H-NMR (400 MHz, CDCl_3_) δ 6.38 (1H, s, 6’-H), 6.37 (1H, s, 7-H), 5.75 (1H, brs, 10-OH), 4.10 (1H, brs, 1-H), 3.77 (1H, brs, 4-H), 3.76 (3H, s, 9-OCH_3_), 3.74 (3H, s, 4’-OCH_3_), 3.61 (1H, d, *J* = 13.5 Hz, 3-*N*-CH_2_Ar), 3.57 (3H, s, 5’-OCH_3_), 3.55 (1H, d, *J* = 13.5 Hz, 3-*N*-CH_2_Ar), 3.38 (3H, s, 2’-OCH_3_), 3.32 (1H, brd, *J* = 7.8 Hz, 5-H), 3.08 (1H, dd, *J* = 18.1, 7.8 Hz, 6-H), 2.99 (1H, dd, *J* = 11.0, 3.0 Hz, 2-H), 2.80 (1H, d, *J* = 11.0 Hz, 2-H), 2.38 (3H, s, ^11^*N* -CH_3_), 2.35 (1H, d, *J* = 18.1 Hz, 6-H), 2.22 (3H, s, 8-CH_3_), 2.13 (3H, s, 3’-CH_3_); ^13^C-NMR (100 MHz, CDCl_3_) δ 151.0 (s, C-2’), 148.9 (s, C-5’), 146.8 (s, C-4’), 145.5 (s, C-10), 142.8 (s, C-9), 130.5 (s, C-6a), 127.9 (s, C-8), 125.5 (s, C-3’), 124.5 (s, C-1’), 120.2 (d, C-7), 119.6 (s, C-10a), 116.7 (s, 4-CN), 109.7 (d, C-6’), 60.9 (q, 9-OCH_3_), 60.6 (q, 4’-OCH_3_), 60.1 (q, 2’-OCH_3_), 58.7 (d, C-4), 55.3 (q, 5’-OCH_3_), 55.3 (d, C-5), 53.4 (t, C-2), 53.3 (t, 3-*N*-CH_2_Ar), 52.4 (d, C-1), 41.4 (q, 11-*N*-CH_3_), 25.0 (t, C-6), 15.5 (q, 8-CH_3_), 9.3 (q, 3’-CH_3_); IR (CHCl_3_) 3534, 3015, 2940, 2226, 1487, 1331, 1227, 1088, 1011 cm^−1^; EIMS *m/z* (%) 467 (M^+^, 1), 441 (13), 440 (48), 247 (16), 246 (12), 245 (57), 205 (19), 204 (100), 195 (20); HREIMS *m/z* 467.2421 (M^+^, calcd for C_26_H_33_N_3_O_5_ 467.2420).

#### 3.1.13. Synthesis of (1*R*,4*R*,5*S*)-9-Methoxy-8,11-dimethyl-7,10-dioxo-3-(2,4,5-trimethoxy-3-methylbenzyl)-1,2,3,4,5,6,7,10-octahydro-1,5-epiminobenzo[d]azocine-4-carbonitrile (**6d**)

To a solution of phenol **25** (16.6 mg, 35.5 µmol) in THF (1 mL) was added salcomine (11.5 mg, 35.5 µmol, 1.0 eq.) at 25 °C, and the reaction mixture was stirred for 3 h under O_2_ atmosphere. The reaction mixture was filtered through a cellulose pad and washed with EtOAc. The filtrate was concentrated in vacuo to give a residue. The residue was purified by SiO_2_ flash column chromatography (n-Hex.−EtOAc = 1:1) to afford compound **6d** (13.5 mg, 79%) as a yellow amorphous. [α]${}_{D}^{27}$ +104.7 (*c* 0.5, CHCl_3_); ^1^H-NMR (400 MHz, CDCl_3_); δ 6.52 (1H, s, 6’-H), 4.01 (3H, s, 9-OCH_3_), 3.88 (1H, brs, 1-H), 3.76 (3H, s, 4’-OCH_3_), 3.69 (1H, brs, 4-H), 3.66 (3H, s, 5’-OCH_3_), 3.61 (2H, s, 3-*N*-CH_2_Ar), 3.56 (3H, s, 2’-OCH_3_), 3.31 (1H, brd, *J* = 7.3 Hz, 5-H), 3.00 (1H, dd, *J* = 11.3, 3.2 Hz, 2-H), 2.69 (1H, dd, *J* = 20.7, 7.3 Hz, 6-H), 2.62 (1H, d, *J* = 11.3 Hz, 2-H), 2.35 (3H, s, 11-*N*-CH_3_), 2.16 (1H, d, *J* = 20.7 Hz, 6-H), 2.15 (3H, s, 3’-CH_3_), 1.94 (3H, s, 8-CH_3_); ^13^C-NMR (100 MHz, CDCl_3_) δ 186.8 (s, C-7), 182.2 (s, C-10), 155.4 (s, C-9), 151.4 (s, C-2’), 149.2 (s, C-5’), 147.5 (s, C-4’), 140.9 (s, C-6a), 137.6 (s, C-10a), 128.3 (s, C-8), 126.0 (s, C-3’), 123.8 (s, C-1’), 116.1 (s, 4-CN), 109.9 (d, C-6’), 61.1 (q, 2’-OCH_3_), 61.0 (q, 9-OCH_3_), 60.2 (q, 4’-OCH_3_), 57.9 (d, C-4), 55.7 (q, 5’-OCH_3_), 54.5 (d, C-5), 53.2 (t, 3-*N*-CH_2_Ar), 51.9 (t, C-2), 51.4 (d, C-1), 41.5 (q, 11-*N*-CH_3_), 20.8 (t, C-6), 9.5 (q, 3’-CH_3_), 8.6 (q, 8-CH_3_); IR (CHCl_3_) 3015, 2941, 2228, 1653, 1308, 1236, 1088, 1009 cm^−1^; EI-MS *m/z* (%) 481 (M^+^, 9), 220 (11), 219 (15), 218 (45), 196 (14), 195 (100); HREIMS *m/z* 481.2212 (M^+^, calcd for C_26_H_31_N_3_O_6_ 481.2213).

#### 3.1.14. Synthesis of 2-(Benzyloxy)-1-(bromomethyl)-3-methoxy-4-methylbenzene (**27**)

To a solution of alcohol **26** (100 mg, 387 µmol) in CH_2_Cl_2_ (2 mL) was added PPh_3_ (125 mg, 465 µmol, 1.2 eq.) and CBr_4_ (162 mg, 465 µmol, 1.2 eq.) at 25 °C, and the reaction mixture was stirred for 6.5 h. The reaction mixture was diluted with H_2_O (10 mL) and extracted with CHCl_3_ (3 × 10 mL). The combined extracts were washed with brine (10 mL), dried over Na_2_SO_4_, and concentrated in vacuo to give a residue. The residue was purified by SiO_2_ flash column chromatography (n-Hex.−EtOAc = 4:1) to afford compound **27** (123 mg, 99%) as a colorless oil. ^1^H-NMR (400 MHz, CDCl_3_) δ 7.54–7.31 (5H, m, Bn-H), 7.03 (1H, d, *J* = 7.8 Hz, 6-H), 6.93 (1H, d, *J* = 7.8 Hz, 5-H), 5.12 (2H, s, 2-OCH_2_Ph), 4.56 (2H, s, 1-CH_2_Br), 3.86 (3H, s, 3-OCH_3_), 2.30 (3H, s, 4-CH_3_); ^13^C-NMR (100 MHz, CDCl_3_) δ 151.7 (s, C-3), 150.1 (s, C-2), 137.4 (s, Bn), 133.6 (s, C-4), 129.9 (s, C-1), 128.5 (d, Bn), 128.4 (d, Bn), 128.2 (d, Bn), 126.1 (d, C-5), 125.2 (d, C-6), 75.2 (t, 2-OCH_2_Ph), 60.2 (q, 3-OCH_3_), 41.4 (t, 1-CH_2_Br), 15.9 (q, 4-CH_3_); IR (CHCl_3_) 3034, 3012, 2936, 1462, 1414, 1278, 1227, 1069 cm^−1^; EIMS *m/z* (%) : 322 (1), 320 (M^+^, 1), 241 (10), 151 (11), 150 (100), 149 (19), 91 (50); HREIMS *m/z* 320.0413 (M^+^, calcd for C_16_H_17_BrO_2_ 320.0412). 

#### 3.1.15. Synthesis of (1*R*,5*S*)-10-(Benzyloxy)-3-(2-(benzyloxy)-3-methoxy-4-methylbenzyl)-9-methoxy-8,11-dimethyl-2,3,5,6-tetrahydro-1,5-epiminobenzo[d]azocin-4(1H)-one (**28**)

To a solution of NaH (60% oil dispersion, 5.70 mg, 142 µmol, 1.0 eq.) in THF (10 mL) was added a solution of lactam **11** (50.0 mg, 142 µmol) in THF (0.7 mL) at 0 °C. The reaction mixture was stirred for 30 min at 0 °C, after which a solution of bromide **27** (45.6 mg, 142 µmol, 1.0 eq.) in THF (0.7 mL) was added at 25 °C. The reaction mixture was stirred for 12 h at 25 °C. The reaction mixture was diluted with H_2_O (20 mL) and extracted with CHCl_3_ (3 × 20 mL). The combined extracts were washed with brine (40 mL), dried over Na_2_SO_4_, and concentrated in vacuo to give a residue. The residue was purified by SiO_2_ flash column chromatography (CHCl_3_−MeOH = 49:1) to afford compound **28** (60.8 mg, 72%) as a colorless oil. [α]${}_{D}^{26}$ −67.0 (*c* 2.2, CHCl_3_); ^1^H-NMR (400 MHz, CDCl_3_); δ 7.37–7.25 (10H, m, 10-*O*-Bn-H, 2’-*O*-Bn-H), 6.71 (1H, s, 7-H), 6.50 (1H, d, *J* = 7.8 Hz, 5’-H), 5.96 (1H, d, *J* = 7.8 Hz, 6’-H), 4.98 (1H, d, *J* = 11.4 Hz, 10-OCH_2_Ph), 4.88 (2H, s, 2’-OCH_2_Ph), 4.84 (1H, d, *J* = 11.4 Hz, 10-OCH_2_Ph), 4.76 (1H, d, *J* = 15.5 Hz, 3-*N*-CH_2_Ar), 4.12 (1H, d, *J* = 15.5 Hz, 3-*N*-CH_2_Ar), 3.90 (1H, brd, *J* = 4.6 Hz, 1-H), 3.76 (3H, s, 3’-OCH_3_), 3.72 (3H, s, 9-OCH_3_), 3.63 (1H, d, *J* = 6.2 Hz, 5-H), 3.59 (1H, dd, *J* = 11.8, 4.6 Hz, 2-H), 3.14 (1H, dd, *J* = 16.9, 6.2 Hz, 6-H), 2.87 (1H, d, *J* = 11.8 Hz, 2-H), 2.82 (1H, d, *J* = 16.9 Hz, 6-H), 2.31 (3H, s, 11-*N*-CH_3_), 2.28 (3H, s, 8-CH_3_), 2.17 (3H, s, 4’-CH_3_); ^13^C-NMR (100 MHz, CDCl_3_) δ 170.3 (s, C-4), 151.2 (s, C-3’), 149.8 (s, C-2’), 149.4 (s, C-9), 148.4 (s, C-10), 137.4 (s×2, Bn), 131.2 (s, C-8), 130.9 (s, C-4’), 128.6 (d, Bn), 128.4 (d×2, Bn), 128.4 (s, C-6a), 128.2 (d, Bn), 128.0 (d, Bn), 127.9 (d, Bn), 126.3 (s, C-10a), 125.7 (d, C-7), 125.7 (d, C-5’), 121.9 (d, C-6’), 74.6 (t, 2’-OCH_2_Ph), 74.1 (t, 10-OCH_2_Ph), 60.0 (q, 3’-OCH_3_), 59.9 (q, 9-OCH_3_), 59.4 (d, C-5), 51.5 (d, C-1), 50.7 (t, C-2), 43.2 (t, 3-*N*-CH_2_Ar), 39.7 (q, 11-*N*-CH_3_), 27.4 (t, C-6), 15.7 (q, 8-CH_3_), 15.6 (q, 4’-CH_3_); IR (CHCl_3_) 3013, 2938, 2467, 1641, 1449, 1337, 1273, 1061 cm^−1^; EIMS *m/z* (%) 593 (17), 592 (M^+^, 40), 295 (26), 294 (100), 204 (29), 203 (20), 91 (10); HREIMS *m/z* 592.2934 (M^+^, calcd for C_37_H_40_N_2_O_5_ 592.2937).

#### 3.1.16. Synthesis of (1*R*,4*R*,5*S*)-10-(Benzyloxy)-3-(2-(benzyloxy)-3-methoxy-4-methylbenzyl)-9-methoxy-8,11-dimethyl-1,2,3,4,5,6-hexahydro-1,5-epiminobenzo[d]azocine-4-carbonitrile (**29**)

To a solution of lactam **28** (85.6 mg, 144 µmol) in THF (4.5 mL) at 0 °C was slowly added LiAlH_2_(OEt)_2_ (1.0 mol/L in CH_2_Cl_2_, 1.70 mL, 1.70 mmol, 12 eq.) over 10 min. The reaction mixture was stirred at 0 °C for 3 h. The reaction mixture was quenched with AcOH (170 µL, 3.00 mmol, 20.8 eq.), followed by the addition of KCN (57.8 mg, 866 µmol, 6.0 eq.) in H_2_O (2.0 mL), and stirring was continued for 14 h at 25 °C. The reaction mixture was neutralized with 5% NaHCO_3_ solution and diluted with saturated Rochell’s salt aq., and the mixture was stirred for 1 h. The reaction mixture was extracted with CHCl_3_ (3 × 30 mL). The combined extracts were washed with brine (40 mL), dried over Na_2_SO_4_, and concentrated in vacuo to give a residue. The residue was purified by SiO_2_ flash column chromatography (n-Hex.−EtOAc = 2:1) to afford compound **29** (61.5 mg, 71%) as a colorless gummy solid. [α]${}_{D}^{27}$ −31.2 (*c* 0.8, CHCl_3_); ^1^H-NMR (400 MHz, CDCl_3_) δ 7.43–7.25 (10H, m, 10-*O*-Bn-H, 2’-*O*-Bn-H), 6.70 (1H, d, *J* = 7.8 Hz, 5’-H), 6.53 (1H, d, *J* = 7.8 Hz, 6’-H), 6.46 (1H, s, 7-H), 5.03 (1H, d, *J* = 11.2 Hz, 10-OCH_2_Ph), 4.72 (1H, d, *J* = 11.2 Hz, 10-OCH_2_Ph), 4.59 (1H, d, *J* = 10.6 Hz 2’-OCH_2_Ph), 4.54 (1H, d, *J* = 10.6 Hz 2’-OCH_2_Ph), 3.95 (1H, brs, 1-H), 3.71 (3H, s, 9-OCH_3_), 3.70 (1H, s, 4-H), 3.68 (3H, s, 3’-OCH_3_), 3.58 (1H, d, *J* = 13.1 Hz, 3-*N*-CH_2_Ar), 3.45 (1H, d, *J* = 13.1 Hz, 3-*N*-CH_2_Ar), 3.23 (1H, brd, *J* = 7.8 Hz, 5-H), 2.95 (1H, dd, *J* = 16.7, 7.8 Hz, 6-H), 2.79 (1H, dd, *J* = 11.0, 3.0 Hz, 2-H), 2.56 (1H, d, *J* = 11.0 Hz, 2-H), 2.27 (1H, d, *J* = 16.7 Hz, 6-H), 2.20 (3H, s, 4’-CH_3_), 2.14 (3H, s, 11-*N*-CH_3_), 2.12 (3H, s, 8-CH_3_); ^13^C-NMR (100 MHz, CDCl_3_) δ 151.8 (s, C-3’), 150.6 (s, C-2’), 148.9 (s, C-9), 148.4 (s, C-10), 137.9 (s, Bn), 137.5 (s, Bn), 131.8 (s, C-4’), 130.1 (s, C-8), 129.9 (s, C-6a), 128.7 (s, C-1’), 128.5 (d, Bn), 128.3 (d, Bn), 128.2 (d, Bn), 128.1 (d, Bn), 127.9 (d, Bn), 127.7 (d, Bn), 126.5 (s, C-10a), 125.5 (d, C-5’), 124.9 (d, C-6’), 124.4 (d, C-7), 116.7 (s, 4-CN), 75.0 (t, 2’-OCH_2_Ph), 74.3 (t, 10-OCH_2_Ph), 60.1 (q, 3’-OCH_3_), 60.0 (q, 9-OCH_3_), 59.4 (d, C-4), 55.4 (d, C-5), 53.8 (t, 3-*N*-CH_2_Ar), 53.7 (t, C-2), 52.7 (d, C-1), 41.2 (q, 11-*N*-CH_3_), 25.0 (t, C-6), 15.7 (q, 8-CH_3_), 15.7 (q, 4’-CH_3_); IR (CHCl_3_) 3015, 2930, 2226, 1454, 1321, 1076, 1028, 700 cm^−1^; EI-MS *m/z* (%) 603 (M^+^, 1), 337 (11), 295 (24), 294 (100), 204 (13), 203 (18), 91 (14); HREIMS *m/z* 603.3099 (M^+^, calcd for C_38_H_41_N_3_O_4_ 603.3097).

#### 3.1.17. Synthesis of (1*R*,4*R*,5*S*)-10-Hydroxy-3-(2-hydroxy-3-methoxy-4-methylbenzyl)-9-methoxy-8,11-dimethyl-1,2,3,4,5,6-hexahydro-1,5-epiminobenzo[d]azocine-4-carbonitrile (**30**)

To a solution of **29** (47.8 mg, 79.2 µmol) and pentamethylbenzene (117 mg, 792 µmol, 10 eq.) in CH_2_Cl_2_ (13 mL) was added BCl_3_ (1.0 mol/L in CH_2_Cl_2_, 400 µL, 400 µmol, 5.0 eq.) over 17 min at −78 °C and the mixture was stirred for 2 h. The reaction mixture was diluted with CH_2_Cl_2_ (10 mL) and quenched with saturated NaHCO_3_ solution (20 mL) at 0 °C. The mixture was extracted with CH_2_Cl_2_ (3 × 20 mL). The combined extracts were dried over Na_2_SO_4_ and concentrated in vacuo to give a residue. To a solution of the obtained residue (168 mg) in THF (5 mL), AcOH (100 µL, 1.66 mmol, 21 eq.) was added. The reaction mixture was stirred for 5 min, after which KCN (31.0 mg, 475 µmol, 6 eq.) in H_2_O (5.0 mL) was added. The reaction mixture was stirred for 1 h at 25 °C. The reaction mixture was neutralized with 5% NaHCO_3_ and diluted with saturated Rochell’s salt aq., and the mixture was stirred for 1 h. The reaction mixture was extracted with CHCl_3_ (3 × 20 mL). The combined extracts were washed with brine (50 mL), dried over Na_2_SO_4_, and concentrated in vacuo to give a residue. The residue was purified by SiO_2_ flash column chromatography (n-Hex.−EtOAc = 2:1) to afford compound **30** (28.5 mg, 85%) as a colorless gummy solid. [α]${}_{D}^{27}$ −46.5 (*c* 0.2, CHCl_3_); ^1^H-NMR (400 MHz, CDCl_3_) δ 7.53 (1H, brs, 2’-OH), 6.67 (1H, d, *J* = 7.8 Hz, 6’-H), 6.58 (1H, d, *J* = 7.8 Hz, 5’-H), 6.52 (1H, s, 7-H), 5.63 (1H, s, 10-OH), 4.16 (1H, brs, 1-H), 3.80 (1H, brs, 4-H), 3.78 (3H, s, 9-OCH_3_), 3.74 (1H, d, *J* = 13.7 Hz, 3-*N*-CH_2_Ar), 3.68 (1H, d, *J* = 13.7 Hz, 3-*N*-CH_2_Ar), 3.67 (3H, s, 3’-OCH_3_), 3.37 (1H, brd, *J* = 7.0 Hz, 5-H), 3.14 (1H, dd, *J* = 19.2, 7.0 Hz, 6-H), 3.01 (1H, dd, *J* = 10.8, 2.7 Hz, 2-H), 2.84 (1H, d, *J* = 10.8 Hz, 2-H), 2.44 (1H, d, *J* = 19.2 Hz, 6-H), 2.41 (3H, s, 11-*N*-CH_3_), 2.29 (3H, s, 8-CH_3_), 2.19 (3H, s, 4’-CH_3_); ^13^C-NMR (100 MHz, CDCl_3_) δ 149.3 (s, C-2’), 146.2 (s, C-3’), 145.4 (s, C-10), 143.3 (s, C-9), 132.0 (s, C-4’), 129.1 (s, C-8), 129.1 (s, C-6a), 123.9 (s, C-6’), 121.4 (d, C-7), 121.3 (d, C-5’), 118.8 (s, C-1’), 118.0 (s, C-10a), 115.3 (s, 4-CN), 60.8 (q, 9-OCH_3_), 59.6 (q, 3’-OCH_3_), 57.9 (d, C-4), 57.8 (t, 3-*N*-CH_2_Ar), 55.1 (d, C-5), 53.3 (t, C-2), 52.2 (d, C-1), 41.5 (q, 11-*N*-CH_3_), 24.7 (t, C-6), 15.9 (q, 8-CH_3_), 15.9 (q, 4’-CH_3_); IR (CHCl_3_) 3532, 3007, 2928, 2232, 1464, 1418, 1242, 1227, 1074 cm^−1^; FABMS *m/z* 424 [M + H]^+^; HRFABMS *m/z* 424.2234 ([M + H]^+^, calcd for C_24_H_30_N_3_O_4_ 424.2236). 

#### 3.1.18. Synthesis of (1*R*,4*R*,5*S*)-9-Methoxy-3-((5-methoxy-4-methyl-3,6-dioxocyclohexa-1,4-dien-1-yl)methyl)-8,11-dimethyl-7,10-dioxo-1,2,3,4,5,6,7,10-octahydro-1,5-epiminobenzo[d]azocine-4-carbonitrile (**6b**) and (1*R*,4*R*,5*S*)-3-(2-hydroxy-3-methoxy-4-methylbenzyl)-9-methoxy-8,11-dimethyl-7,10-dioxo-1,2,3,4,5,6,7,10-octahydro-1,5-epiminobenzo[d]azocine-4-carbonitrile (**6c**)

To a solution of phenol **30** (17.3 mg, 40.8 µmol) in THF (1.5 mL) was added salcomine (27.6 mg, 81.6 µmol, 2.0 eq.) at rt, and the reaction mixture was stirred for 18 h under O_2_ atmosphere. The reaction mixture was filtered through a cellulose pad and washed with EtOAc. The filtrate was concentrated in vacuo to give a residue. The residue was purified by SiO_2_ flash column chromatography (n-Hex.−EtOAc = 2:1) to afford compound **6b** (8.0 mg, 43%) as a yellow oil, and **6c** (5.6 mg, 31%) as a yellow oil.

**6b**: [α]${}_{D}^{27}$ −70.3 (*c* 0.3, CHCl_3_); ^1^H-NMR (400 MHz, CDCl_3_) δ 6.27 (1H, t, *J* = 1.8 Hz, 2’-H), 4.00 (3H, s, 9-OCH_3_), 3.90 (3H, s, 5’-OCH_3_), 3.86 (1H, brs, 1-H), 3.74 (1H, brd, *J* = 1.8 Hz, 4-H), 3.49 (1H, d, *J* = 16.5, 1.8 Hz, 3-*N*-CH_2_Ar), 3.39 (1H, d, *J* = 16.5, 1.8 Hz, 3-*N*-CH_2_Ar), 3.34 (1H, brd, *J* = 7.3 Hz, 5-H), 2.98 (1H, dd, *J* = 11.1, 3.0 Hz, 2-H), 2.75 (1H, dd, *J* = 20.7, 7.3 Hz, 6-H), 2.54 (1H, d, *J* = 11.1 Hz, 2-H), 2.34 (3H, s, 11-*N*-CH_3_), 2.20 (1H, d, *J* = 20.7 Hz, 6-H), 1.98 (3H, s, 8-CH_3_), 1.89 (3H, s, 4’-CH_3_); ^13^C-NMR (100 MHz, CDCl_3_) δ 187.4 (s, C-3’), 186.7 (s, C-7), 182.5 (s, C-6’), 182.2 (s, C-10), 155.9 (s, C-5’), 155.5 (s, C-9), 141.3 (s, C-1’), 140.9 (s, C-6a), 136.9 (s, C-10a), 132.9 (d, C-2’), 129.1 (s, C-8), 129.0 (s, C-4’), 116.0 (s, 4-CN), 61.1 (q, 9-OCH_3_), 60.7 (q, 5’-OCH_3_), 59.3 (d, 4-C), 54.7 (d, 5-C), 52.3 (t, 3-*N*-CH_2_Ar), 51.2 (d, 1-C), 50.8 (t, 2-C), 41.4 (q, 11-*N*-CH_3_), 20.8 (t, 6-C), 8.7 (q, 8-CH_3_), 8.5 (q, 4’-CH_3_); IR (CHCl_3_) 3017, 2945, 2359, 2230, 1655, 1612, 1308, 1234, 1153 cm^−1^; EIMS *m/z* (%) 451 (M^+^, 6), 261 (18), 260 (37), 233 (11), 232 (25), 220 (12), 219 (43), 218 (100), 204 (26), 190 (11), 176 (13), 166 (19), 83 (10); HREIMS *m/z* 451.1740 (M^+^, calcd for C_24_H_25_N_3_O_6_ 451.1743).

**6c**: [α]${}_{D}^{27}$ +95.3 (*c* 0.2, CHCl_3_); ^1^H-NMR (400 MHz, CDCl_3_) δ 7.92 (1H, brs, 2’-OH), 6.68 (1H, d, *J* = 7.8 Hz, 6’-H), 6.60 (1H, d, *J* = 7.8 Hz, 5’-H), 4.00 (3H, s, 9-OCH_3_), 3.92 (1H, brs, 1-H), 3.78 (1H, d, *J* = 14.0 Hz, 3-*N*-CH_2_Ar), 3.75 (1H, brs, 4-H), 3.73 (1H, d, *J* = 14.0 Hz, 3-*N*-CH_2_Ar), 3.70 (3H, s, 3’-OCH_3_), 3.38 (1H, brd, *J* = 7.4 Hz, 5-H), 2.99 (1H, dd, *J* = 11.7, 3.2 Hz, 2-H), 2.79 (1H, dd, *J* = 20.8, 7.4 Hz, 6-H), 2.70 (1H, d, *J* = 11.7 Hz, 2-H), 2.36 (3H, s, 11-*N*-CH_3_), 2.23 (1H, d, *J* = 20.8 Hz, 6-H), 2.20 (3H, s, 4’-CH_3_), 2.00 (3H, s, 8-CH_3_); ^13^C-NMR (100 MHz, CDCl_3_) δ 186.2 (s, C-7), 182.1 (s, C-10), 155.4 (s, C-9), 149.0 (s, C-2’), 146.1 (s, C-3’), 140.8 (s, C-6a), 136.9 (s, C-10a), 132.2 (s, C-4’), 129.2 (s, C-8), 124.0 (d, C-6’), 121.6 (d, C-5’), 118.2 (s, C-1’), 114.9 (s, 4-CN), 61.0 (q, 9-OCH_3_), 59.8 (q, 3’-OCH_3_), 57.5 (d, C-4), 57.2 (t, 3-*N*-CH_2_Ar), 54.3 (d, C-5), 51.3 (t, C-2), 51.1 (d, C-1), 41.5 (q, 11-*N*-CH_3_), 20.7 (t, C-6), 15.9 (q, 4’-CH_3_), 8.8 (q, 8-CH_3_); IR (CHCl_3_) 3524, 3022, 2945, 2853, 2359, 2234, 1653, 1614, 1308, 1236, 1152 cm^−1^; EIMS *m/z* (%) : 437 (M^+^, 5), 411 (24), 410 (100), 261 (19), 260 (80), 259 (12), 245 (12), 234 (24), 233 (20), 232 (43), 231 (14), 220 (21), 219 (49), 218 (98), 217 (12), 204 (26), 203 (13), 202 (15), 192 (19), 190 (12), 176 (13), 151 (14), 150 (33), 149 (17), 91 (13), 77 (12). HREIMS *m/z* 437.1956 (M^+^, calcd for C_24_H_27_N_3_O_5_ 437.1951).

### 3.2. Biological Evaluation

A single-cell suspension of each cell line (2 × 10^3^ cells/well) was added to the serially diluted test compounds in a 96-well microplate and cultured for 4 days. Cell viability was measured with Cell Counting Kit-8 (Dojindo Laboratories, Kumamoto, Japan). IC_50_ was expressed as the concentration at which cell growth was inhibited by 50% compared with the untreated control.

## 4. Conclusions

We presented a short and efficient methodology for the preparation of the chiral right-half model compounds of renieramycins. The synthesized model compounds were screened for their cytotoxic activity against DU145 and HCT116. Compounds **6a** and **21** bearing benzyl group at 3-nitrogen showed very strong activity with IC_50_ at nanomolar concentrations. It was also found that chirality had no effect on the cytotoxic activities of the model compounds.

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
