# Peer review of "Asymmetric Synthesis and Cytotoxicity Evaluation of Right-Half Models of Antitumor Renieramycin Marine Natural Products"

_marinedrugs, 2018, doi:10.3390/md17010003_

Round 1
Reviewer 1 Report
This is a good paper in which the synthesis and the cytotoxic activity of some analogs of the right-hand part of renieramycins are presented.
The authors have a considerable experience in this field as witnessed by the various papers previously published dealing with the same subject.
The paper is well written though in some parts English can be a little improved. The chemistry developed is sound and overall the paper deserves publication on Marine drugs after minor revisions. A careful analysis of the text to eliminate typos is required.
For example line 172: Chemistry and not Chemictry.
Author Response
Thanks for your reflection and giving us a nice evaluation.
A careful analysis of the text to eliminate typos is required.
For example line 172: Chemistry and not Chmictry.
This manuscript ha been pre-checked by a native speaker and finally confirmed yypos carefuly, before submission (see, attached certifile file).
Reviewer 2 Report
The authors report on the asymmetric synthesis and cytotoxicity evaluation of molecules related to the right-half structure of renieramycin. The work was carried out in a rigorous way and the biological values are relevant. Only minor comments are given, according to the indication below.
A more appropriate form must be found for 3N-;
For the molecules 6b and 6d in Figure 3 it is better indicate the common structure with a R group on N position and then indicate the corresponding unit for the two products
It is not correct introduce 6d before 6c in the text.
In the caption of Scheme 1 add “[9,11]” after 6a.
In the caption of figure 4, replace CDE model with “CDE-ring model“
At page 5, line 114, replace benzyl bromide 22 with “substituted benzyl bromide 22”
In scheme 4, the molecular structure of 22 ,without brackets, must be moved closer to 22, under 55%
At page 6, line 146, remove reference [21] and introduce the corresponding text in Experimental section as a new paragraph 3.2, biological evaluation.
At page 8, paragraph 3.1.2, replace DMSO with “DMSO-d6” for 1H and 13CNMR analysis
In reference[1] remove capital letters in the title
In reference [21] remove J.Nat. Prod. and indicate only submitted; the journal is correctly indicated if the work is accepted.
